# Extraction of Essential Oil from River Tea Tree (*Melaleuca bracteata* F. Muell.): Antioxidant and Antimicrobial Properties

**Mursleen Yasin** [1,*], **Adnan Younis** [1], **Fahad Ramzan** [1], **Talha Javed** [2,3], **Rubab Shabbir** [2], **Hamza Armghan Noushahi** [4], **Milan Skalicky** [5], **Peter Ondrisik** [6], **Marian Brestic** [6], **Sabry Hassan** [7] and **Ayman EL Sabagh** [8,*]

1. Institute of Horticultural Sciences, University of Agriculture Faisalabad, Faisalabad 38040, Pakistan; adnanyounis@uaf.edu.pk (A.Y.); fahidflorist@yahoo.com (F.R.)
2. College of Agriculture, Fujian Agriculture and Forestry University, Fuzhou 350002, China; mtahaj@fafu.edu.cn (T.J.); rubabshabbir28@gmail.com (R.S.)
3. Department of Agronomy, University of Agriculture Faisalabad, Faisalabad 38040, Pakistan
4. College of Plant Science and Technology, Huazhong Agricultural University, Wuhan 430070, China; hamzanaushahi143@gmail.com
5. Department of Botany and Plant Physiology, Faculty of Agrobiology, Food and Natural Resources, Czech University of Life Sciences Prague, 16 500 Prague, Czech Republic; skalicky@af.czu.cz
6. Department of Plant Physiology, Slovak University of Agriculture, 949 01 Nitra, Slovakia; peter.ondrisik@uniag.sk (P.O.); marian.brestic@uniag.sk (M.B.)
7. Department of Biology, College of Science, Taif University, P.O. Box 11099, Taif 21944, Saudi Arabia; sabryh@lycos.com
8. Department of Agronomy, Faculty of Agriculture, University of Kafrelsheikh, Kafrelsheikh 33516, Egypt
* Correspondence: mursleenyasin@gmail.com (M.Y.); ayman.elsabagh@agr.kfs.edu.eg (A.E.S.)

**Abstract:** Tea tree oil (TTO) from the genus *Melaleuca* L. has antimicrobial, antibacterial, antifungal, and antioxidant properties and is used by the cosmetic, pharmaceutical, and horticultural industries. In Pakistan, *Melaleuca bracteata* can be exploited for essential oil purposes, as this species is well adapted to Pakistan's agroclimatic conditions. The objective of the present study was to evaluate the yield of *M. bracteata* essential oil together with its antioxidant and antimicrobial properties under local prevailing conditions of the subtropics. Essential oil was extracted through the hydrodistillation method. Using this method, six batches of 8 kg samples (fresh leaves and branches) underwent a distillation process for 4–5 h. The average yield obtained was about 0.2%. The GCMS was used to identify the components of extracted essential oil. Eugenol methyl ether is the major component in extracted essential oil, i.e., 96% of the total. A high content of flavonoids and phenolics and a Fe-reducing power ability of *M. bracteata* were observed. The oil was also found effective against *B. subtilis*, *B. cereus*, White rot, and *A. flavus*. Hence, it is concluded that there is a possibility to use TTO for its biocidal properties, and it must also be inspected and then commercialized in Pakistan by the agriculture and cosmetic industries.

**Keywords:** *Melaleuca bracteata*; essential oil; hydrodistillation; eugenol methyl ether; antioxidant; antimicrobial

## 1. Introduction

Natural products from plants are under immense consideration in the organic world. One such utilization is of essential oils that are complex volatile compounds extracted from leaves, twigs, flowers, seeds, branches, buds, fruits, and roots. These secondary compounds are then stored in secretory cells, glandular trichomes, canals, cavities, and epidermal cells [1]. Use of essential oils (EOs) is expeditiously increased because of their antibacterial, antiviral, antifungal, insecticidal [1], anti-inflammatory [2], and antimicrobial traits [3]. EOs also have antidepressant and anxiolytic activities [4]. These characters are possessed by certain chemical compounds, mainly phenolics, alkaloids, flavonoids, glycosides, and flavonoids [5].

The Myrtaceae family would rank among the most important families of plants which are enriched with essential oils. Genus Melaleuca is one of the prominent members of the family Myrtaceae that is also well known for its essential oils. *Melaleuca* is indigenous to Australia yet can adjust into several agroclimatic conditions. The volatile oil product of this genus is familiar to the world with the name of tea tree oil (TTO). The species that are under commercial use for essential oil production are *Melaleuca alternifolia*, *M. cajuputi*, *M. bracteata*, and *M. quinquenervia* [6]. Plant extract including oil was used by Aboriginals and was also utilized by soldiers as a general antimicrobial and insect repellent in World War II [7]. Today, TTO is gaining popularity in different industries and being used in aromatherapy, herbal and allopathic medicines, cosmetics, and as natural biocide, fungicide, and preservative. The oil is slightly yellow colored to transparent with a strong camphor-like odor and menthol-like cooling effect [8,9]. The composition of TTO has almost 100 different chemicals, predominantly monoterpenes (terpinen-4-ol, terpinolene, p-cymene, α-pinene, γ-terpinene, 1,8-cineole), sesquiterpenes, and their respective alcohols (monoterpene, alcohol-terpineol) [10].

In Pakistan, the *Melaleuca* genus has been cultivated in the Punjab region for a long time, but its properties and products are yet not investigated here. *Melaleuca bracteata* is one of the species that is well adapted to the agroclimatic conditions of Pakistan [11]. Tea tree oil contains antioxidant, cytotoxic, antifungal, insecticidal, and medicinal properties [12–14]. Reports have shown methyl eugenol and phenylpropanoids to be the prime components of tea tree oil from *M. bracteata*. Moreover, it can be inferred from literature that TTO can be used effectively by the horticulture sector as a chemical-free fungicide, bactericide, and for enhancement of the shelf life and vase life of fruits, vegetables, and flowers. Hence, it is necessary to test the effectiveness and appropriate use of this essential oil so it can be extracted and employed by the local industry more cost-effectively. Therefore, this study aims to investigate the local adapted species of tea tree oil for more exploration and assessment of this essential oil.

## 2. Materials and Methods

### 2.1. Plant Material

Fresh plant material including leaves and twigs were collected from the University of Agriculture Faisalabad and Qadir Baksh Nursery Farms Faisalabad, Pakistan (31.43° N, 73.07° E). The sample for extraction was prepared by clipping leaves and young twigs into smaller pieces.

### 2.2. Extraction Method

Essential oil was extracted from Melaleuca via the hydrodistillation technique at the Department of Chemistry, University of Agriculture Faisalabad, Pakistan (31.43° N, 73.07° E). The unit consisted of a distillation flask/tank, condenser, and thermometer. About 8–10 kg of plant material was added in flask and submerged into 12 L of water. Then, the set up was airtightly closed, and the process was started at 250 °C temperature. The procedure continued for 4–5 h. Oil was collected in a separating funnel. The %age yield of the extracted oil ($v/w$) was calculated by the following formula [15], and then the oil was stored at a cool dry place [16].

$$\text{Essential Oil Yield \% = Volume of extracted oil/Fresh weight of plant material (g)} \times 100. \qquad (1)$$

### 2.3. Analysis of Tea Tree oil by GCMS

The extracted oil samples were subjected to compound analysis via the gas chromatography–mass spectrometer technique. The detector of Agilent 5975C was used, and it was provisioned with 7890A GC and column of HP-5MS with 30 m length, 250 micrometer width, and 0.25 micrometer internal diameter with a temperature of 450°C. The oven program was kept on, and the equilibrium time for the oven was 0 min. The program was run for 36 min, and the temperature of the heater was 240 °C. The inert gas

used as a carrier was Helium with a flow rate of 0.7 mL/min. Mass spectrometry was performed in EMV mode at relative voltage of 59eV and scan range from 50 to 550amu. The mass spectrometer was equipped with a gold standard quadrupole analyzer at 150 °C (max 200 °C). The temperature of MS source was 240 °C (max 250 °C). The Agilent MSD Productivity ChemStation6 was the software which identified the compounds receiving signals at the same time from both detectors [15].

### 2.4. Antioxidant Analysis

### 2.4.1. DPPH Radical Scavenging Activity

Radical scavenging activity is the ability of an antioxidant compound to protect cells from free radicals produced during normal cell metabolism. To check this character of tea tree oil, DPPH (2,2-diphenyl-1-picrylhydrazyl) scavenging activity was carried out. For this purpose, 1.0 mL at 0.3 M of 1,1-diphenyl-2-picrylhydrazyl was added to the 2.5 mL solution of the sample and gallic acid standard. Standards of gallic acid were prepared at 20, 40, 60, 80, and 100 ppm. Then, all the test tubes were incubated at room temperature for 20 min and kept in the dark. After that, absorbance was recorded at 518 nm by spectrophotometer (T60 U Spectrophotometer, PG Instruments Ltd., Leicestershire, UK). DPPH 100 ppm was used as control treatment, and absorbance was recorded as Abs control. A blank experiment of 80% ethanol was run as well, and absorbance was calculated as Abs blank. The DPPH scavenging activity was determined from the following expression [17].

$$\text{DPPH scavenging activity (\%)} = 100 - \{[(\text{Abs sample-Abs blank}) \times 100]/\text{Abs control}]\}$$

### 2.4.2. Total Flavonoid Contents

Flavonoid compounds are antioxidants present in plants and all fruits and vegetables. To evaluate this in tea tree oil, 1.0 mL of samples and catechin standards at 20, 40, 60, 80, and 100 ppm were added in 4.0 mL of distilled water. The solution was prepared in a volumetric flask of 10 mL capacity. After this, 5% $NaNO_2$ weighing 0.3 mL was mixed in the above prepared solution. 1 M NaOH (2 mL) was added after 1 min, and later, after 5 m, 0.3 mL of 10% $AlCl_3$ was also mixed. After 60 s, 2 mL of 1 M NaOH was added to make the total volume up to 10 mL by deionized distilled water. We mixed it thoroughly, and absorbance was observed at 510 nm using reagent blank. Catechin equivalent in milligram per gram dry plant material was used to present the total flavonoids in TTO [18].

### 2.4.3. Total Phenolic Compounds

To estimate the phenolic compounds of tea tree oil, gallic acid standards at 20, 40, 60, 80, and 100 ppm were used. To 1.0 mL volume of both samples and standards, 5.0 mL of Folin-Ciocalteu (Fisher chemicals) and 4 mL of $Na_2CO_3$ (7% $w/v$) were added with continuous shaking. Then, all the solutions were kept in dark for 30 min, and later on, absorbance was detected at 765 nm with the help of a spectrophotometer (T60 U Spectrophotometer, PG Instruments Ltd., Leicestershire, UK). A blank sample used was a reagent solution. A gallic acid equivalent (GAE) was used to express the amount of total phenolics based on milligram per gram of plant dry weight [19].

### 2.4.4. Reducing Power Ability (RPA) of the Plant

It is determined by adding 1.0 mL gallic acid standard solutions and sample to 0.2 molar solution of phosphate buffer solution (2.3 mL) at 6.6 pH. Then, 2.5 mL of 1% potassium ferricyanide ($K_3[Fe(CN)_6]$) was added to the solution. Incubation of the mixture was done at 37 °C for 20 m. Then, 10% trichloroacetic acid (2.5 mL) was mixed and then centrifuged at 1000 rpm for 10 m. The supernatant with a 2.5 mL volume was mixed with distilled water and 0.5 mL of 0.1% $FeCl_3$. After 10 m, absorbance was determined at 700 nm using a spectrophotometer (T60 U Spectrophotometer, PG Instruments Ltd., Leicestershire, UK) [20].

## 2.5. Antimicrobial Analysis

The inhibitory zone (IZ) assay of tea tree oil was tested on microbes using the disc diffusion method as proposed by Al-Abd et al. [21]. Firstly, an agar plate was prepared by spreading the microbe, then paper disks of antimicrobials were added. This method is mostly used to determine the best antibiotic against a new or drug-resistant pathogen. The zone of inhibitions was measured with the help of a transparent scale in millimeters [22]. The bacterial strains tested by this procedure were *Bacillus subtilis* and *Bacillus cereus*. The fungal strains evaluated against tea tree oil were White rot and *Aspergillus flavus*.

## 2.6. Statistical Analysis

The lab experiments were conducted through a completely randomized design (CRD). Data were analyzed using *Statistix 8.1* software (https://www.statistix.com/ (accessed on: 9 February 2020)), and mean analysis was conducted through Tukey's honestly significant difference test on a 0.05 level of probability [23].

## 3. Results and Discussion

### 3.1. Essential Oil Yield

Yield of essential oil via the hydrodistillation method was about 0.2%, as shown in Table 1. The volume of extracted oil could vary based on the agroclimatic and geographical conditions of the area. A study by Siddique et al. [24] reported the %age yield of *Melaleuca bracteata* to range at $0.14 \pm 0.01\%$ in Pakistan. The percentage yield of extracted oil of each hydrodistillation batch is shown in Table 1.

**Table 1.** %age Yield of tea tree oil from *M. bracteata* leaves.

| Batch No. | % Age Yield |
|:---:|:---:|
| 1 | $0.25 \pm 0.017$ |
| 2 | $0.2 \pm 0.017$ |
| 3 | $0.15 \pm 0.015$ |

### 3.2. GCMS Analysis

The GCMS analysis of tea tree oil from *M. bracteata* showed the 7 peaks as shown in Figure 1. Table 2 indicates that methyl eugenol is the supreme compound of tea tree oil extracted from *Melaleuca bracteata* species. It represents 96% of the total, and the other six contents were found as just trace elements. The other elements in order of percentage are p-Mentha-2,8-diene-1-ol, Methyl cinnamate, Germacrene D, Terpinolene, Linalool, and alpha-phellandrene. Previous literature has also proclaimed methyl eugenol as the major component of *M. bracteata* oil, with 86.5% of the total volume [24,25]. The latter also indicated methyl cinnamate as the second highest compound, representing 11% of the total. However, another team of scientists documented the percentage of methyl eugenol to be greater than 95% in the tea tree oil of *bracteata*, which is quite related to the present finding [26].

**Table 2.** GC–MS analysis of tea tree oil from *Melaleuca bracteata*.

| Peak No. | Identified Compounds | Retention Time (min) | % of Total |
|:---:|:---:|:---:|:---:|
| 1 | alpha-phellendrene | 8.568 | 0.49% |
| 2 | p-Mentha-2,8-diene-1-ol | 9.278 | 0.92% |
| 3 | Terpinolene | 11.071 | 0.64% |
| 4 | Linalool | 11.595 | 0.50% |
| 5 | Methyl cinnamate | 19.359 | 0.77% |
| 6 | Methyl eugenol | 19.897 | 96.02% |
| 7 | Germacrene D | 20.663 | 0.67% |

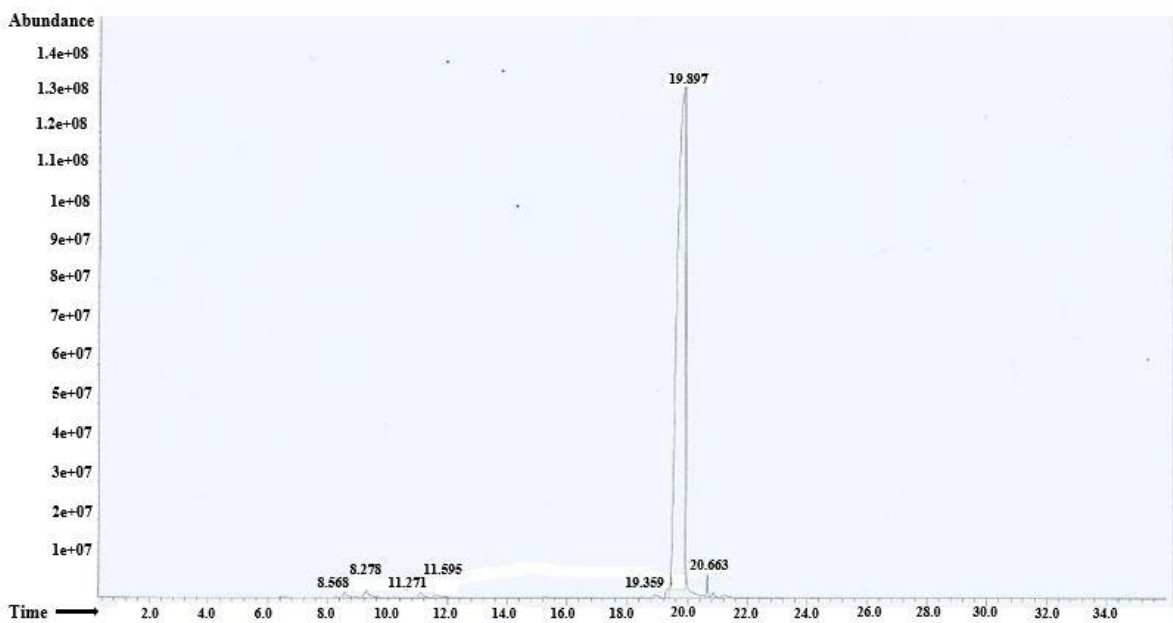

**Figure 1.** Chromatogram of TTO by GCMS analysis.

Methyl eugenol (ME) belongs to phenylpropanoids, and its occurrence is found in 450 species of plants. In nature, ME is known for plant chemical defense as an antifungal, antibacterial, antinematode and has toxic activities against insect pathogens [27]. The oil from *Echinophora sibthorpiana* (Apiaceae) also has methyl eugenol. Kivanc [28] found that its oil (~0.1%) and ME at 0.05-0.1 % have negative effects on the growth of fungi and bacteria. Later, Kivanc and Akgul [29] identified that eugenol methyl ether at 1000 ppm also detained mycelium and spore formation of *Aspergillus* and *Penicillium* strains. It has also been found to inhibit fungal infestation in peanuts by Sudhakar et al. [30]. ME has been shown to act as a floral synomone in the coevolution of orchid species in the genus *Bulbophyllum* with fruit flies. It was recognized as the best insect repellent and most effective in knockdown and killing effects among the major constituents of EOs [31]. The importance of methyl eugenol can also be inferred from the findings of Rossi et al. [32] that volatile oil from carrot was successfully evaluated to inhibit the growth of *Campylobacter jejuni*, a major gastroenteritis-causing bacterium across the globe. Methyl eugenol, a component of carrot oil, was also found effective separately at a minimum inhibitory concentration of 250 µg/mL.

### 3.3. Biochemical Assays

The oil extracted from *Melaleuca* species exhibited competent antioxidant properties. The result is supported by Table 3, which shows the antioxidant values of *M. bracteata* oil along with the mean and standard error values.

**Table 3.** Average antioxidant values of TTO from *Melaleuca bracteata*.

| Replication No. | TFC Value (mm/g) | TPC Value (mm/g) | RPA Value (mm/g) | DPPH %Age |
|---|---|---|---|---|
| R1 | 955.67 | 108.945 | 300.79 | 89.13 |
| R2 | 942.16 | 108.945 | 314.88 | 86.95 |
| R3 | 955.67 | 112.145 | 307.81 | 84.45 |
| Mean | 951.16 | 110.01 | 307.83 | 86.84 |
| S.D | 7.80 | 1.84 | 7.04 | 4.22 |
| S.E | 4.50 | 2.60 | 4.06 | 2.43 |

TFC: total flavonoid contents, TPC: total phenolic compounds, RPA: reducing power ability, DPPH: 2,2-diphenyl-1-picrylhydrazyl.

### 3.3.1. DPPH Scavenging Activity

In Table 3, the average scavenging activity of tea tree oil is displayed. DPPH values for *M. bracteata* oil ranged at 86.848 ± 2.43 mg/mL from the three replicates.

### 3.3.2. Total Flavonoid Contents

From Table 3, it is inferred that TTO has a relatively higher amount of flavonoids, i.e., 951 ± 4.50 mg/mL. Flavonoids are produced in plants as a response to oxidative pressure and photodamage via secondary metabolic processes [33,34].

### 3.3.3. Total Phenolic Compounds of Tea Tree Oil

The TPC of tea tree oil were exploited with respect to the gallic acid standard. The slope expression obtained from the standard curve was used to calculate the concentration of phenolics as a gallic acid equivalent (GAE) in mg/g of plant dry weight basis. Table 3 represents a higher concentration of phenolics, i.e., 110 mg/g of the dry weight of *Melaleuca bracteata* plant.

### 3.3.4. Reducing Power Ability

Antioxidants reduce $Fe^{3+}$ to $Fe^2$, i.e., ferricyanide complex to the ferrous form. This transformation is observed at 700 nm [35]. The reducing ability is amplified with increased absorbance [36]. RPA values were evaluated as equivalent to the standard. About 308 mg/g of the dry weight of the plant was found in *Melaleuca bracteata* oil. A study by Hou et al. [37] provided evidence regarding the antioxidant potential of *Melaleuca bracteata*. When extraction was done with conventional methods, the total phenolic compounds found were 88.6 ± 1.3 mg, and the total flavonoids were 19.4 ± 0.2 mg. The DPPH scavenging activity was high at 86 ± 0.3%. After conditions were optimized, all the values of antioxidants were heightened and found to be at their peak values of 98.7 ± 1.2 mg, 21.6 ± 0.3 mg, and 94.7 ± 0.8% for TPC, TFC, and DPPH, respectively. Siddique et al. [24] also reported that scavenging activity in *M. bracteata* ranged between 35.3 and 89.2 ± 0.4%. This potential increased with the increase of concentration of oil from 20 to 100 μg/mL.

Tea tree oil is an effective antioxidant agent as its commercial species, i.e., *Melaleuca alternifolia*, has also been observed to be enriched with this potential by several reports. One such is the study of Noumi et al. [38] in which DPPH scavenging activity was recorded at 12.5 μg/mL expressed as IC50 values (the concentration of antioxidant at which 50% of the reaction was inhibited). The power of TTO as an antioxidant was also found to be better than that of common synthetic antioxidant butylated hydroxytoluene (BHT) at 30 mM. Previous researchers proposed TTO as a good alternative to maintain the oxidative stability of the food matrix. Additionally, when compared with other natural oxidants such as vitamin C, vitamin E, and quercetin, *Melaleuca alternifolia* oil expressed strong free radical scavenging properties and inhibited lipid peroxidation [39]. This is because of the presence of phenols in this oil, which reduce the aerobic oxidation of organic matter [40]. Hence, it is concluded that the present detection of antioxidants in tea tree oil from *Melaleuca bracteata* is in accordance with previous reports of several scientific studies, and this species could also be explored as an antioxidative essential oil along with its patent plant species (*Melaleuca alternifolia*).

### 3.4. Antimicrobial Properties

*Bacillus subtilis* and *Bacillus cereus* were tested against tea tree oil from *Melaleuca bracteata*. The zone of inhibition was measured in millimeters for all treatments. It is the measure of a clear region with no microbial growth around the paper disc on the agar surface. The clear area represents the resistance or the effective inhibition of microbial growth by antimicrobial agent. The larger the region is, the greater the antimicrobial effect of the agent. the effect of tea tree oil on the growth of *Bacillus subtilis* and *Bacillus cereus* is elaborated on in Figure 2. The zone of inhibition obtained by the application of *M. bracteata*

oil was 19.5 and 13 mm $\pm$ 0.43. The oil was efficient in controlling bacterial growth better than standard treatment.

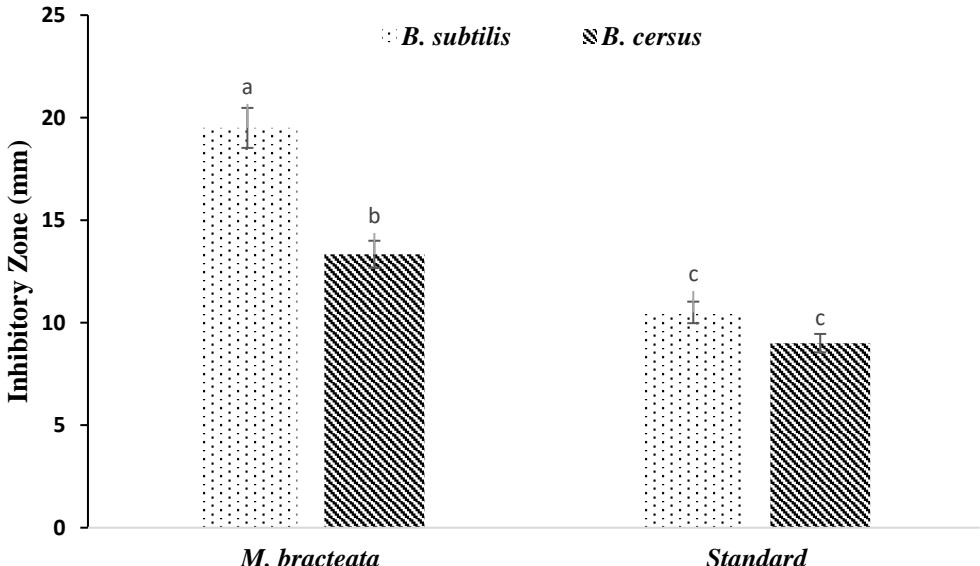

**Figure 2.** Zone of inhibition calculated from *M. bracteata* oil sample and standard against *Bacillus* species.

*Melaleuca* species possess good antibacterial effects against *Bacillus* strains. Previously, in a finding by Goswami et al. [41], the zone of inhibition for *B. subtilis* by *M. bracteata* oil was measured to be 6–7 mm, which showed a moderate antifungal effect of essential oil on the bacterial strain. The growth of *B. subtilis* subsp. *Spizizenii* was also inhibited by this oil species at 250 µg/mL up to 44 mm zone of inhibition [24]. EO from this species was also found to be effective against *Staphylococcus aureus* strains, *S. epidermidis*, *Klebsiella pneumonia*, and *Streptococcus mutans* when tested by Goswami et al. (2017) [41]. The MIC of *M. alternifolia* oil counted for *Bacillus cereus* was 0.3% (*v/v*) by Griffin et al. [42]. Previous literature reported that TTO has a bactericidal activity against *Staphylococcus* species, *Lactobacillus*, *Actinomyces viscosus*, and *E. coli* [43–46]. The antimicrobial traits of *M. alternifolia* have been employed to its terpinen-4-ol compound [47,48]. Moreover, terpinen-4-ol and methyl eugenol from both the species have been recognized to have efficient antibacterial, antifungal, and antinematode properties [14,27].

### 3.5. Antifungal Properties

The potential of fungicidal effects of tea tree oil was evaluated by measuring its zone of inhibition for *Aspergillus flavus* and *Phanerochaete chrysosporium* (White rot). From Figure 3, it is inferred that the growth of White rot fungi and *A. flavus* was restricted up to 11.67 mm and 10.5 mm by the application of oil from *M. bracteata* species, respectively.

Shin [49] has documented the antifungal aspects of commercial TTO from *M. alternifolia* against *Aspergillus flavus*. The concentration required for the effective control on fungus ranged between 0.3 and 0.7 (% volume by volume). The oil has been found to be effective against other *Aspergillus* species such as *A. niger* with a minimum inhibitory concentration of 0.016 to 0.4 (%volume/volume). The effective antifungal agent is terpinen-4-ol in this oil. Moreover, the zone of inhibition formed by methyl eugenol from the oil of another *Melaleuca* species was 24.3 $\pm$ 0.3 mm for pure oil and 46 $\pm$ 0.6 mm at 100 µg/mL of oil against *Fusarium oxysporum* [50]. Homeyer et al. [51] observed that within 15 min of exposure, TTO reduced the log of *Fusarium oxysporum* up to -4 as compared to control treatment.

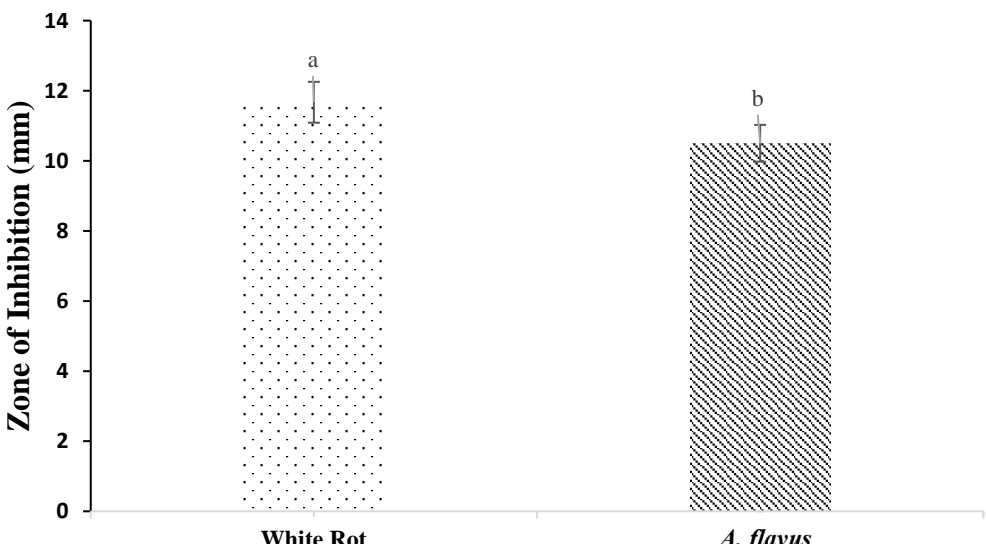

**Figure 3.** Zone of inhibition calculated from *M. bracteata* oil against White Rot and *Aspergillus flavus*.

## 4. Conclusions

The present study aimed at the extraction of essential oil from *Melaleuca bracteata* to explore its antioxidant and antimicrobial properties. It has been found from the results that the yield percentage for essential oil is about 2% through the hydrodistillation method, and it is enriched with eugenol methyl ether. A high antioxidant activity is observed when measured on scales for total flavonoids, total phenolics, reducing power ability, and DPPH. The laboratory application of TTO on microbes reduces their activity. Thus, tea tree oil from *Melaleuca bracteata* species has significant potential as an antioxidant and antimicrobial to be used as a natural microbicide in the food and agriculture industries. The *Bracteata* species is well adapted to Pakistan and specifically the Punjab region's climate. Thus, it must be exploited by the local industry to produce raw and refined oil products to commercialize its use in horticulture, pharmaceuticals, and cosmetics.

**Author Contributions:** Conceptualization, A.Y.; methodology, M.Y.; software, M.Y.; validation, T.J., R.S., F.R. and A.E.S.; formal analysis, M.Y.; investigation, M.Y.; resources, A.Y.; data curation, T.J., R.S. and A.E.S.; writing—original draft preparation, M.Y.; writing—review and editing, M.S., M.B., P.O., A.Y., T.J., H.A.N. and A.E.S.; supervision, A.Y.; project administration, A.Y.; funding acquisition, S.H., A.Y. and A.E.S. All authors have read and agreed to the published version of the manuscript.

**Funding:** The authors was funded by Taif University Researchers Supporting Project grant number (TURSP-2020/142), Saudi Arabia.

**Acknowledgments:** The authors are highly grateful to Taif University Researchers Supporting Project Number (TURSP-2020/142), Saudi Arabia. The authors thank Muhammad Zaryab Khalid for critically reviewing the manuscript.

**Conflicts of Interest:** The authors declare no conflict of interest.

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
