# Peer review of "Extraction of Essential Oil from River Tea Tree (Melaleuca bracteata F. Muell.): Antioxidant and Antimicrobial Properties"

_sustainability, doi:10.3390/su13094827_

Round 1

Reviewer 1 Report

The manuscript titled Extraction of essential oil from Melaleuca bracteata to study its antioxidant and antimicrobial properties deals with the evaluation of its performance and its application in local subtropical conditions. 
The paper is interesting but needs some correction:

Line 17: Please remove the hyphen in the word, it should be Republic
Line 19: the author's full address is missing

Introduction: I suggest rewording the introduction slightly, especially:
Line 31: no explanation of the abbreviation TTO used, it only appears in line 49, this should be changed
Line 42: no explanation of the abbreviation EOs
Line 83: the equation and its description should be refined
Line 88: the name of the technique should be written in small letters
Line 95: lack of information about other parameters of the method, no indication of the source of the methodology used
Line 107: refine the presented equation and its description
Line 132: the names of the chemical compounds in the sentence should be written in small letters
Line 157: add refinement to the presented equation and its description
Line 166: trace elements or analytes?
Line 185: Bulbophyllum write in italica font
Line 200: no explanation of abbreviations used in the table 
Line 278: Melaleuca write in italica font

Line 281: Figure 4- A.flavus write in italica font

The conclusions thoroughly supported by the results presented in the article, but very laconically.

Author Response

Dear Reviewer

               Thank you for your valuable suggestion and comments for the improvement of current manuscript. We have improved the manuscript based your comments. All the mentioned changes have been incorporated in the manuscript. We appreciate for your warm work earnestly and hope that the correction will meet with approval.

On behalf of all co-authors, once again thank you for your valuable efforts.                                                        

Best regards,

Ayman El Sabagh

Response to Reviewer

The manuscript titled Extraction of essential oil from Melaleuca bracteata to study its antioxidant and antimicrobial properties deals with the evaluation of its performance and its application in local subtropical conditions. 
The paper is interesting but needs some correction:

Comment 1: Please remove the hyphen in the word, it should be Republic.

Author’s response: Hyphen is removed according to your suggestion.

Comment 2: Line 19: the author's full address is missing

Author’s response: Esteemed reviewer, thank you for highlighting an important mistake. We have provided the complete address.

Comment 3: Introduction: I suggest rewording the introduction slightly, especially: no explanation of the abbreviation TTO used, it only appears in line 49, this should be changed

Author’s response: The explanation of abbreviation is added where the word is used first time. Moreover, we have carefully reviewed the Introduction section and rewarding is done according to your suggestion.

Comment 4: no explanation of the abbreviation Eos

Author’s response: Explanation of the abbreviation EOs is added in the manuscript according to your suggestion.

Comment 5: the equation and its description should be refined

Author’s response: The equation is refined and description is added under heading 3.2.

Comment 6: the name of the technique should be written in small letters

Author’s response: It is changed to small letters according to your suggestion.

Comment 7: lack of information about other parameters of the method, no indication of the source of the methodology used

Author’s response: More information about parameters is added and reference is also mentioned at the end of paragraph now.

Comment 8: refine the presented equation and its description

Author’s response: The description of equation is mentioned in the relevant paragraph (above the equation).

Comment 9: the names of the chemical compounds in the sentence should be written in small letters

Author’s response: The names of the chemical compounds have been written in small letters accordingly.

Comment 10: add refinement to the presented equation and its description

Author’s response: The equation description is added in paragraph 3.2 and removed from here because of the repetition of same equation.

Comment 11: trace elements or analytes?

Author’s response: The compounds identified in GCMS analysis as less than 1% were referred as trace elements.

Comment 12: Bulbophyllum write in italica font

Author’s response: Bulbophyllum has been written in italic font

Comment 13: no explanation of abbreviations used in the table 

Author’s response: Explanation is added in the last row of the table

Comment 14: Melaleuca write in italica font

Author’s response: Changed accordingly.

Comment 15: Figure 4- A.flavus write in italica font

Author’s response: A. flavus has been written in italic font according to your suggestion.

Comment 16: The conclusions thoroughly supported by the results presented in the article, but very laconically.

Author’s response: Conclusion section has been improved according to your suggestion.

Reviewer 2 Report

Overall the manuscript addresses an interesting topic and prior to publication some improvements  are required.

The English language needs improvements, the sound is not scientific enough and many spell check and rephrasing paragraph is required. A correction by an experimented speaker and a more scientific writing style is mandatory.

The abstract needs to be restructured because the aim and conclusions of the research are not clear presented. Mandatory the conclusions must be improved and correlated with all analyzes performed. Also, in the introductory part it is mandatory to reformulate the purpose to be more concise and clearer.  A more detailed justification for the research aim would be of greater relevance.

The methods are described with sufficient details to allow another researcher to reproduce the experiments, but the conclusions need major improvements. Also, please provide a better-quality version for figure 1.

Author Response

Dear Reviewer

               Thank you for your valuable suggestion and comments for the improvement of current manuscript. We have improved the manuscript based your comments. All the mentioned changes have been incorporated in the manuscript. We appreciate for your warm work earnestly and hope that the correction will meet with approval.

On behalf of all co-authors, once again thank you for your valuable efforts.           

Best regards,

Ayman El Sabagh

Response to Reviewer

Overall the manuscript addresses an interesting topic and prior to publication some improvements are required.

Comment 1: The English language needs improvements, the sound is not scientific enough and many spell check and rephrasing paragraph is required. A correction by an experimented speaker and a more scientific writing style is mandatory.

Author’s response: Esteemed reviewer, we have tried our best to improve the manuscript based on your suggestion. Moreover, Dr. Zaryab Khalid (South China Agricultural University, China) has thoroughly reviewed the manuscript for English editing.

Comment 2: The abstract needs to be restructured because the aim and conclusions of the research are not clear presented. Mandatory the conclusions must be improved and correlated with all analyzes performed. Also, in the introductory part it is mandatory to reformulate the purpose to be more concise and clearer.  A more detailed justification for the research aim would be of greater relevance.

Author Response: The changes have been made according to your suggestion. A more defined aim and objective for the study has been added to the abstract and introduction as well. Conclusion is improved and correlated with the results section.

Comment 3: The methods are described with sufficient details to allow another researcher to reproduce the experiments, but the conclusions need major improvements. Also, please provide a better-quality version for figure 1.

Author Response: Esteemed reviewer, we have improved the conclusion and correlated it with the results section. The quality of figure 1 is improved according to your suggestion.

Reviewer 3 Report

The manuscript cannot be considered suitable for Sustainability. The topic, as the authors themselves declare in the meagre introduction, is not innovative and numerous and more detailed papers are already available in the international literature. The objective of the work itself is not clear: the authors declare that they want to “investigate the locally adapted species of tea tree oil”: but what do they do to achieve this? In the materials and methods some sentences are typical of the introduction (e.g. lines 109-110) and in the results, the same formula (lines 157-160) of materials and methods (lines 83-86) is reported. The results and the discussion (and it could not have been otherwise given the meagre information obtained from the experimental activity) are decidedly not very thorough. Reporting the same data, both as a table and as a figure (table 3 and figure 2), only worsens the judgment on the manuscript. Therefore, I do not believe that the manuscript can be considered suitable for publication.

Author Response

Dear Reviewer

               Thank you for your valuable suggestion and comments for the improvement of current manuscript. We have improved the manuscript based your comments. All the mentioned changes have been incorporated in the manuscript. We appreciate for your warm work earnestly and hope that the correction will meet with approval.

On behalf of all co-authors, once again thank you for your valuable efforts.                                                        

Best regards,

Ayman El Sabagh

Response to Reviewer

Comment: The manuscript cannot be considered suitable for Sustainability. The topic, as the authors themselves declare in the meagre introduction, is not innovative and numerous and more detailed papers are already available in the international literature. The objective of the work itself is not clear: the authors declare that they want to “investigate the locally adapted species of tea tree oil”: but what do they do to achieve this? In the materials and methods some sentences are typical of the introduction (e.g. lines 109-110) and in the results, the same formula (lines 157-160) of materials and methods (lines 83-86) is reported. The results and the discussion (and it could not have been otherwise given the meagre information obtained from the experimental activity) are decidedly not very thorough. Reporting the same data, both as a table and as a figure (table 3 and figure 2), only worsens the judgment on the manuscript. Therefore, I do not believe that the manuscript can be considered suitable for publication.

Author’s response: Esteemed reviewer, thank you for your valuable suggestions. In Pakistan, the current study was executed for the very first time. The study is important because there is a possibility to use TTO for its biocidal properties and must also be inspected and then commercialized in Pakistan by agriculture and cosmetic industry. Presently, the methods and chemicals used as for achieving are antimicrobial properties are neither economical nor environmental friendly. Therefore, the authors think that the current study has greater significance under the climatic conditions of Pakistan. Dear reviewer, we are sorry for inconvenience, we have clearly explained the objective of current study in abstract as well as in Introduction section according to your suggestion. We have thoroughly improved the introduction section. The repeated formula has been removed. The results section is thoroughly improved. We have removed the figure 2 from manuscript and revised the results section in more acceptable form. We appreciate your suggestion and believe that now the corrections according to your suggestions will meet with approval.

Round 2

Reviewer 2 Report

In general, the article has been improved, but I would still recommend a more careful spelling correction. 

Author Response

Dear Reviewer

            Thank you for your valuable suggestion and comments for the improvement of current manuscript. We have improved the manuscript based on your comments. All the mentioned changes have been incorporated in the manuscript. We appreciate for your warm work earnestly and hope that the correction will meet with approval.

On behalf of all co-authors, once again thank you for your valuable efforts.                                              

Best regards,

Ayman El Sabagh

Response to Comment

Comment: In general, the article has been improved, but I would sill recommend a more careful spelling correction.

Author’s response: Esteemed reviewer, we have carefully checked the whole manuscript for any spelling mistake. All the mentioned changes have been incorporated in the manuscript. Corrections have been done according to your suggestion.

Reviewer 3 Report

Refusing a manuscript is certainly not easy, because I realize that numerous authors have contributed to its writing and that the manuscript itself is the results of a commitment that always deserves respect. Nevertheless, the few changes made do not change my negative opinion of the manuscript. Beyond the characterization of the oil, if this product is so important to Pakistan, many more investigations than those presented would have been needed. I am therefore forced to reiterate, despite myself, the initial judgment.

Author Response

Dear Reviewer

            Thank you for your valuable suggestion and comments for the improvement of current manuscript. We have improved the manuscript based on your comments. All the mentioned changes have been incorporated in the manuscript. We appreciate for your warm work earnestly and hope that the correction will meet with approval.

On behalf of all co-authors, once again thank you for your valuable efforts.                                            

Best regards,

Ayman El Sabagh

Revision Round 2

Response to Comment

Comment: Refusing a manuscript is certainly not easy, because I realize that numerous authors have contributed to its writing and that the manuscript itself is the results of a commitment that always deserves respect. Nevertheless, the few changes made do not change my negative opinion of the manuscript. Beyond the characterization of the oil, if this product is so important to Pakistan, many more investigations than those presented would have been needed. I am therefore forced to reiterate, despite myself, the initial judgment.

Author’s response: Thank you for your valuable suggestion and comments for the improvement of current manuscript. During the process of revising our manuscript, we do appreciate what you have done and the constructive and valuable comments.

Revision Round 1

Response to Reviewer

Comment: The manuscript cannot be considered suitable for Sustainability. The topic, as the authors themselves declare in the meagre introduction, is not innovative and numerous and more detailed papers are already available in the international literature. The objective of the work itself is not clear: the authors declare that they want to “investigate the locally adapted species of tea tree oil”: but what do they do to achieve this? In the materials and methods some sentences are typical of the introduction (e.g. lines 109-110) and in the results, the same formula (lines 157-160) of materials and methods (lines 83-86) is reported. The results and the discussion (and it could not have been otherwise given the meagre information obtained from the experimental activity) are decidedly not very thorough. Reporting the same data, both as a table and as a figure (table 3 and figure 2), only worsens the judgment on the manuscript. Therefore, I do not believe that the manuscript can be considered suitable for publication.

Author’s response: Esteemed reviewer, thank you for your valuable suggestions. In Pakistan, the current study was executed for the very first time. The study is important because there is a possibility to use TTO for its biocidal properties and must also be inspected and then commercialized in Pakistan by agriculture and cosmetic industry. Presently, the methods and chemicals used as for achieving are antimicrobial properties are neither economical nor environmental friendly. Therefore, the authors think that the current study has greater significance under the climatic conditions of Pakistan. Dear reviewer, we are sorry for inconvenience, we have clearly explained the objective of current study in abstract as well as in Introduction section according to your suggestion. We have thoroughly improved the introduction section. The repeated formula has been removed. The results section is thoroughly improved. We have removed the figure 2 from manuscript and revised the results section in more acceptable form. We appreciate your suggestion and believe that now the corrections according to your suggestions will meet with approval.
